# The Thermoelectric Properties of Spongy PEDOT Films and 3D-Nanonetworks by Electropolymerization

**DOI:** 10.3390/nano12244430

**Published:** 2022-12-12

**Authors:** Cristina V. Manzano, Olga Caballero-Calero, Aída Serrano, Pedro M. Resende, Marisol Martín-González

**Affiliations:** 1Instituto de Micro y Nanotecnología, IMN-CNM, CSIC (CEI UAM+CSIC), Isaac Newton 8, E-28760 Tres Cantos, Spain; 2Departamento de Electrocerámica, Instituto de Cerámica y Vidrio, CSIC, Kelsen 5, E-28049 Madrid, Spain

**Keywords:** PEDOT, films, 3D nanonetworks, electropolymerization, thermoelectric properties

## Abstract

Recently, polymers have been attracted great attention because of their thermoelectric materials’ excellent mechanical properties, specifically their cost-effectiveness and scalability at the industrial level. In this study, the electropolymerization conditions (applied potential and deposition time) of PEDOT films were investigated to improve their thermoelectric properties. The morphology and Raman spectroscopy of the PEDOT films were analyzed according to their applied potential and deposition time. The best thermoelectric properties were found in films grown at 1.3 V for 10 min, with an electrical conductivity of 158 ± 8 S/cm, a Seebeck coefficient of 33 ± 1 µV/K, and a power factor of 17 ± 2 µW/K·m^2^. This power factor value is three times higher than the value reported in the literature for electropolymerized PEDOT films in acetonitrile using lithium perchlorate as a counter-ion. The thermal conductivity was found to be (1.3 ± 0.3) × 10^−1^ W/m·K. The highest figure of merit obtained at room temperature was (3.9 ± 1.0) × 10^−2^ using lithium perchlorate as a counter-ion. In addition, three-dimensional (3D) PEDOT nanonetworks were electropolymerized inside 3D anodic aluminum oxide (3D AAO), obtaining lower values in their thermoelectric properties.

## 1. Introduction

Because of the necessity to build new thermoelectric devices and convert them into flexible parts, organic materials such as polymers have been used as an alternative because of their excellent mechanical properties, specifically their nontoxicity, price, and scalability in the industry. Different semiconductor polymers were studied for their thermoelectric properties, such as polyaniline [1], PCDTBT [2], polyacetylene [3], etc. One of the most studied polymers for such applications is PEDOT (poly(3,4-ethylenedioxythiophene)) [4,5,6]. The most common growth techniques to obtain PEDOT films are drop coating [4], spin coating [5,6,7,8], gel film [9], and electropolymerization [10,11,12].

PEDOT:PSS (poly(3,4-ethylenedioxythiophene)-polystyrene sulfonate) is the most studied form of PEDOT in thermoelectric applications [13]. This is because, compared with other organic materials, it offers high thermoelectric performance. Normally, these films are obtained using a commercial suspension of PEDOT:PSS, which is then deposited by drop coating or spin coating. To improve the thermoelectric properties of films, different solvents were used, including DMSO (dimethyl sulfoxide) [4], ethylene glycol [4], formic acid [5], sulfuric acid [8], deionized water, isopropanol, or acetone [6]. In addition, the thermoelectric properties were also improved by different posttreatments, such as obtaining a hybrid material with tellurium [8] or measuring via temperature [6,7], as shown in Table 1. The values of the power factor obtained in these studies were around 50 [6,8], 80 [5] or 95 [7] µW/m·K^2^. The highest power factor, 470 µW/m·K^2^, with a *zT* of 0.42, was claimed by Kim et al. while using ethylene glycol [4] as a solvent.

Electropolymerization is an excellent growth method thanks to its high control over the structural and morphological properties of the materials. In the literature, several studies can be found on the fabrication of PEDOT films by electropolymerization [14,15,16,17,18]. However, only a few studies have been published covering both the growth of PEDOT films by electropolymerization and their transport or thermoelectric properties. In 2014, Castagnola et al. [10] reported the effect of the different electrochemical routes on the morphology and electrical conductivity of the films by using an aqueous solution of EDOT (ethylenedioxythiophene) and NaPSS. The electrical conductivity increased when the films were grown in the potentiodynamic mode compared with the potentiostatic mode, and the lowest value was found for the galvanostatic mode. Culebras et al. [11] also published a paper in 2014 on the effect of counter-ions (ClO_4_, PF_6_ and BTFMSI (bis(trifluoromethylsulfonyl)imide)) on the thermoelectric properties of the polymer. The solvent used was acetonitrile and the electropolymerization conditions were the same (galvanostatic mode at −3 mA vs. Ag/AgCl for 2 min), independent of the counter-ion used. The highest thermoelectric figure of merit of 0.22 at room temperature (RT), with a power factor of 147 µW/m·K^2^ and a thermal conductivity of 0.19 ± 0.02 W/m·K, was obtained when BTFMSI, which is very expensive, was used as a counter-ion and reduced with hydrazine after the growth. A comparison of thermoelectric properties, on the basis of the solvents (acetonitrile and a mixture of water and methanol) used for electropolymerization and film thicknesses, was published in 2019 [12]. The best power factor, of 41.3 µW/m·K^2^, was measured when water and methanol mixtures were used as the electrolyte and galvanostatic mode was also applied. Because different electrical conductivities were obtained, depending on the electropolymerization potential [10], it is essential to study the effects of the applied potential on thermoelectric properties.

Furthermore, as proven in inorganic materials such as bismuth telluride [19,20], the thermoelectric properties can be enhanced by reducing thermal conductivity through nanostructuration. In the literature, nanowires embedded in PEDOT films grown by chemical oxidation [21,22,23], one-dimensional (1D) and three-dimensional (3D) [24] PEDOT nanowires obtained by electropolymerization inside the lithographic substrate [25], mesoporous silica [26], polycarbonate membranes [27], or anodic aluminum oxide templates [28] can be found. With regard to the thermoelectric properties of these nanostructures, the highest power factor was 92 µW/m·K^2^ for PEDOT nanowires electropolymerized inside a lithographic substrate.

In this work, PEDOT films were obtained by electropolymerization in the potentiostatic mode, changing the deposition conditions of applied potential and electropolymerization time. The influence of the growth conditions on the chemical structure and morphological properties of the films was studied. Moreover, the thermoelectric properties of the PEDOT films grown under these different conditions, their applied potential, and the electrodeposition time were analyzed. Three-dimensional (3D) PEDOT nanostructures were also electropolymerized inside the 3D anodic aluminum oxide (3D AAO), and their thermoelectric properties were evaluated. In our work, the influence of the potentiostatic mode in electropolymerized PEDOT films on thermoelectric properties was studied for the first time. Therefore, this work opens a new way to improve the thermoelectric properties of PEDOT films by using a growth technique that is cost-effective and industrially scalable and that can be applied to the development of flexible thermoelectric devices.

## 2. Experimental Methods

### 2.1. Fabrication Method of Electropolymerized PEDOT Films

The fabrication of the films was performed by electropolymerization using a solution in acetonitrile (99.8% from Sigma Aldrich, Darmstadt, Germany) of 0.01 M EDOT (97% from Sigma Aldrich) and 0.1 M LiClO_4_ (99.99% from Sigma Aldrich), following what has already been reported [11,12]. Lithium perchlorate was used as a counter-ion for the electropolymerization of EDOT in PEDOT. Electropolymerization was carried out using a conventional three-electrode electrochemical cell at RT controlled by an Autolab PGSTAT101 bi-potentiostat. A Pt mesh, Ag/AgCl, and 150 nm Au/5 nm Cr/Si were used as the counter, reference, and working electrodes, respectively. In addition, to obtain 3D PEDOT nanonetworks, 3D AAO templates were prepared by applying a pulsed current-density method. In this process, a periodic current-density profile was applied to generate a modulated layered structure. The combination of this approach with the acidic nature of the electrolyte led to a 3D architecture, presenting both longitudinal (nanopores) and transversal connections. A detailed description of this method, including Al (advent research materials, 99.999%) substrate cleaning and polishing, is available in [29,30]. After the pulsed profile, an added layer of standard nanopores was grown, conferring additional mechanical support and allowing for the posterior separation of the template from the aluminum substrate and its preparation for electropolymerization. The anodization was performed with a 1.1 M sulfuric acid electrolyte (25% *v*/*v* ethanol), at −1 °C. After anodization, the AAO template was cleaned with deionized water and then dried. Kapton tape was used to clean the template surface, exfoliating the overetched upper layers. The aluminum substrate was chemically etched with a CuCl_2_/HCl aqueous solution, after which the AAO barrier layer was removed with an aqueous phosphoric acid solution at 30 °C. Before electropolymerization, 150 nm Au/5 nm Cr layers were deposited on the AAO side that had the 3D AAO structure, ensuring electric contact but also limiting the polymerization in the 3D zone of the template. The electropolymerization inside the 3D AAO was carried out in similar conditions to those for the PEDOT films.

### 2.2. Morphological Characterization and Chemical Structure of PEDOT Films

The morphology of the films and nanostructures was analyzed using field emission-scanning electron microscopy (FE-SEM, FEI VERIOS 460, Thermo Fisher Scientific, Waltham, MA, USA) with a 2 kV accelerating voltage. The thickness of the films was measured using a stylus profiler system (Vecco^®^ Dektak, Bruker, Billerica, MA, USA). Raman spectroscopy was performed to analyze the vibrational modes of PEDOT films prepared to vary the deposition conditions: applied potential and electropolymerization time. These measurements were performed using a high-resolution Raman spectrometer (Horiba Jobin Yvon, Kioto, Japan) with a 532 m Nd:YAG laser (8.5 mW) from 200 to 1700 cm^−1^ in air at RT.

### 2.3. Thermoelectric Characterization of PEDOT Films

To measure the thermoelectric properties of PEDOT films, it was necessary to detach the films from the conductive substrate, which in our case was a Si substrate coated with gold and chromium layers. The detachment process consisted of immersing the films in ethanol to separate them from the conductive substrate, and then the film was transferred to a glass substrate. The electrical conductivity was measured using an Ecopia Hall measurement system, Toronto, ON, Canada (HMS-5500) at RT. The Seebeck coefficient was measured using a lab-made system at RT by applying different temperature gradients between 1 °C and 5 °C. The slope between the generated Seebeck voltage and the temperature gradient gave the Seebeck coefficient. Because of the measurement systems used, both electrical conductivity and Seebeck coefficient were obtained in-plane. The errors of the electrical conductivity and Seebeck coefficient were 5% and 10%, respectively. The thermoelectric power factor was calculated by using the electrical conductivity and the Seebeck coefficient in the in-plane direction. Finally, the thermal conductivity was measured by the photoacoustic method [2,19,31,32,33,34,35]. The photoacoustic (PA) technique was calibrated, and it was explained in the supporting information of some of the mentioned works. This method involved the periodic illumination of the samples’ surface, once it has been coated with a 80 nm titanium layer, with a modulated laser (in our case, a fiber coupled from Alphalas of 980 nm in wavelength and a maximum intensity of 260 mW). The absorption of the light produced periodic heating in the sample, creating acoustic waves due to the heating of the surrounding air. These acoustic waves were detected by a microphone (40 BL 1/4′′ CCP pressure type, with a 26 CB, 1/4′′ preamplifier, both from G.R.A.S. Sound & Vibration). The layer’s thermal diffusivity was calculated by using a multilayer model developed by Hu et al. [36], which used the phase shift between the incident light and the recorded sound. A reference sample consisting of a quartz substrate was also used. The thermal conductivity, *k*, can then be calculated using the equation *k* = *α*·*ρ*·*Cp*, where *ρ* is the theoretical density and *Cp* is the specific heat of the layer of interes. In this case, the density was 1.47 g/cm^3^ [37], and the specific heat was 0.95 J/g.K [38]. The error rate associated with the thermal conductivity obtained by the photoacoustic technique was approximately 20%.

## 3. Results and Discussion

### 3.1. Fabrication of PEDOT Films

Cyclic voltammetry (see Figure 1) was performed from the open-circuit potential (OCP) of −0.25 V toward the oxidation state (1.8 V), then to a reduction state (−1.3 V), and finally back to the OCP vs. Ag/AgCl with a scan rate of 0.01 V/s to determine the onset oxidation potential and the appropriate potential range for the electropolymerization of PEDOT on the gold substrate to fabricate PEDOT films. The solution used was 0.01 M EDOT and 0.1 M LiClO_4_, which was a supporting electrode, in acetonitrile.

The onset potential was taken at the intersection of the tangents drawn at the baseline current and the oxidation current slope in the cyclic voltammetry. In our case, the onset potential was observed at 1.15 V. Another important feature was the crossover between the forward and reverse scans. This is called the “nucleation loop” and is attributed to the initial stage of nucleation processes for conductive polymer films. In our case, this peak was found at ~1.3 V. The oxidation region went from 1.15 V to 1.4 V. In this region, the EDOT monomer was oxidized, becoming a polymer through a diffusion process. These EDOT monomers were oxidized to radical cations and then dimerized and deprotonated. After this step, the dimer was oxidized, and the formation of oligomeric radical cation species occurred. This oligomer bound with other EDOT^•+^, forming the PEDOT polymer [12,14]. The oxidation of EDOT is an irreversible process up to 2 V [39], and therefore, in this study, the cyclic voltammetry was performed up to 1.8 V. In addition, a reduction peak was observed at −0.5 V (see Figure 1), showing that the electropolymerization process of PEDOT was reversible.

The selection of the potential to perform the electropolymerization is important in the growth of PEDOT films, and it can affect the structural and morphological properties of the films and, consequently, their thermoelectric properties. According to the cyclic voltammetry, the potential region of interest ranged from 1.15 V vs. Ag/AgCl to 1.5 V vs. Ag/AgCl. Therefore, the electropolymerization potentials of 1.3 V vs. Ag/AgCl and 1.4 V vs. Ag/AgCl were selected as the electropolymerization potentials in the first stage of the nucleation processes and the final stage of the oxidation region, respectively, to grow and fine-tune the structural, morphological, and thermoelectric properties of PEDOT films. In addition, three electropolymerization times (5, 10, and 15 min) were also studied to determine the influence of the film thicknesses on the thermoelectric properties. Additionally, when applying lower potentials, such as 1.2 V vs. Ag/AgCl, no film was obtained, and for higher potentials, such as 1.5 V vs. Ag/AgCl, the transport properties were too low to be considered of interest.

### 3.2. Morphological Characterization and Chemical Structure of PEDOT Films

The influence of the oxidation potential and polymerization time on the morphological properties of the films was analyzed using FE-SEM. Figure 2 shows the top view of FE-SEM images for the different electropolymerization conditions. The top view at higher magnification is an inset in all FE-SEM images.

A comparison of the PEDOT films obtained with the two applied potentials (1.3 V and 1.4 V) shows that their morphologies are different. The morphology observed for the PEDOT films grown at 1.3 V presents a similar morphology to the cauliflower structure. The size of these cauliflowers increases when the deposition time increases. When the electropolymerization time is increased from 5 min to 15 min, the cauliflower size increases from 1 µm to 2 µm. The morphologies observed in the films deposited at 1.4 V with 5 min and 15 min of deposition times are similar, where the respective sizes increase as the deposition times increase. In this case, the morphology resembles a mesh of small cauliflowers, whereas the films grown at 1.4 V for 10 min have a slightly different morphology, a network-like morphology. This morphology is like the morphology observed in previous works [11,12] where PEDOT films were grown by electropolymerization.

The film thickness is an important parameter for electrical conductivity, which was measured using a profilometer to get the exact value. The thicknesses for the film deposited at 1.3 V were found to be 4.1, 6.0, and 7.3 µm for electropolymerization times of 5, 10, and 15 min, respectively. For the films electropolymerized at 1.4 V, the thicknesses were 4.3, 10.4, and 23.2 µm for 5, 10, and 15 min deposited times, respectively. As expected, the films became thicker as the electropolymerization time increased.

The vibrational modes of PEDOT films were studied using Raman spectroscopy. Figure 3 shows the Raman spectra of the films performed under an applied potential of 1.3 V and 1.4 V and different electropolymerization times, of 5, 10, and 15 min. As reported in [40], all the peaks can be identified with the vibrational modes of PEDOT, in all the films. An assignment of the active Raman bands is shown in Figure 3, according to [38]. The Raman modes found at 440, 579, 855, and 989 cm^−1^ were due to oxyethylene ring deformation. Symmetric C-S-C deformation was found at 706 cm^−1^, C-O-C deformation was observed at 1129 cm^−1^, the peak at 1262 cm^−1^ was identified as C_α_-C_α_ (inter-ring) deformation, C_β_-C_β_ stretching was located at 1367 cm^−1^, symmetric C_α_ = C_β_ (-O) stretching was found at 1445 cm^−1^, and asymmetric stretching of C = C appeared at 1505 and 1571 cm^−1^.

To compare the different spongy PEDOT films grown in this study, the Raman spectra were normalized to the most intense band, the one corresponding to the symmetric C_α_ = C_β_ (-O) stretching at 1445 cm^−1^. The position, the intensity, and the full width half maximum (FWHM) of the different Raman vibration modes for the PEDOT films were determined by Lorentzian functions, and they are shown in Table 2. As shown in Figure 3 and Table 2, similar Raman spectra were obtained for all the PEDOT films, regardless of the electropolymerization potential and growth time, identifying all the Raman bands associated with PEDOT. However, changes in position, FWHM, and relative intensity were identified depending on the electropolymerization conditions.

In general, with respect to the Raman position of the bands, a blue shift is observed for most Raman modes as electropolymerization time increases (see Figure 3 and Table 2), independent of the applied potential. This behavior may be associated with the chemical bond length of molecules [41] due to changes in the structures induced by the growth conditions. The greater Raman shift is observed toward lower wavenumbers in the symmetric C_α_ = C_β_ (-O) stretching band of the film prepared at 1.4 V for 5 min. According to Mengistie et al. [5], the red shift of this vibrational mode could indicate that the chain in the resonant structure of PEDOT changes from a benzoid to a quinoid structure [42]. This change in the PEDOT structure increases electrical conductivity, but in our case, this change into quinoid was not enough to modify the whole polymer structure (the Raman shift is very low). Therefore, the electrical conductivity should not be affected by this structural change. Concerning the FWHM of the Raman bands, a trend was not noted, and variations may relate to modifications in the structures and structural defects generated during the electropolymerization. The strongest changes in the Raman spectra of PEDOT films were shown in the relative intensity of the Raman shifts between 1300 and 1600 cm^−1^ (Figure 3). In this region, the double bonds of PEDOT are found, and for thicker films (structures prepared at higher electropolymerization times), a higher intensity of the signal of these Raman modes was observed. This was due to the neutral PEDOT segments’ being more active for a green laser than the doped segments were [11].

### 3.3. Thermoelectric Characterization of PEDOT Films

The figure of merit of the thermoelectric materials depends on the Seebeck coefficient (*S*), electrical conductivity (*σ*), thermal conductivity (*κ*), and absolute temperature, *T*, as shown in Equation (1):(1)$$zT=\frac{\sigma \cdot S^{2}}{\kappa }\cdot T$$

The power factor is given by the electrical conductivity times the Seebeck coefficient to the power of two. Figure 4 depicts the transport properties (*S* and *σ*) of electropolymerized PEDOT films at various applied potentials and deposition times.

The electrical conductivity increased as the electropolymerization times increased (see Figure 4A) for the films grown at 1.3 V. The values obtained were 109 ± 6, 158 ± 8, and 174 ± 9 S/cm for 5, 10, and 15 min, respectively, while for those grown at 1.4 V, the conductivity was observed at half of that obtained at 1.3 V and decreased even more with the deposition time. The values of the electrical conductivity of the films obtained at 1.4 V were 57 ± 3, 49 ± 3, and 28 ± 1 S/cm for 5, 10, and 15 min, respectively. The difference in electrical conductivity between the films obtained at 1.3 V and at 1.4 V can be explained by the red shift observed in the Raman spectrum. As was already explained, the red shift of the symmetric C_α_ = C_β_ (-O) stretching Raman band indicates a change in the PEDOT structure from a benzoid to a quinoid structure [42] (see Figure 5).

The highest electrical conductivity was measured on the PEDOT film electropolymerized at 1.3 V for 15 min, where the benzoid structure was maximized. For the electrical conductivity, the value found in the literature for films grown in acetonitrile and lithium perchlorate was 200 S/cm [11,12], and the value found in films electropolymerized using Na:PSS in water was 150 S/cm [10]. Then, the maximum electrical conductivity obtained in this study was of the same order of magnitude as those of the values found in the literature.

The Seebeck coefficients were positive in all the films measured, showing that PEDOT is a *p-type* semiconductor (see Figure 4B). This magnitude decreased as the deposition time increased, except for the film grown at 1.4 V for 5 min. The Seebeck coefficients for the films grown at 1.3 V were found to be 31 ± 2, 33 ± 1, and 28 ± 1 µV/K for 5, 10, and 15 min, respectively. The values of this magnitude for the films deposited at 1.4 V were 27 ± 2, 21 ± 2, and 13 ± 4 µV/K for 5, 10, and 15 min, respectively. Finally, the Seebeck coefficient was higher for films deposited at 1.3 volts, with the highest value discovered in films deposited for 10 min. The Seebeck coefficients found in the literature for films grown under similar conditions (acetonitrile, lithium perchlorate, and electropolymerization) were 9 µV/K [11] and 18.9 µV/K [12]. The PEDOT film grown in this work with the highest Seebeck coefficient exhibits a value 1.8 times higher than the highest reported.

Because of the values obtained for the electrical conductivity and the Seebeck coefficient, the highest power factor of 17 ± 2 µW/K·m^2^ can be calculated for film grown at 1.3 V for 10 min. The power factor value obtained in this study is more than two times higher than that obtained in previous studies [11,12]. Then, we showed that by studying only the electropolymerization parameters, it was possible to improve the power factor of PEDOT films without using a post-treatment after the growth.

According to the SEM images, the spongy PEDOT film presents a porous structure. That is something that must be considered when studying thermal conductivity. In this work, the photoacoustic (PA) technique was calibrated and used to extract the thermal conductivity of the film at RT, as we have done in previous works [2,19,31,32,33,34,35]. The PA measurements can be perfectly fitted with a film composed of 90% of polymer and 10% air. Then, the thermal conductivity values of the spongy film are 0.19 ± 0.04, 0.13 ± 0.03, and 0.15 ± 0.03 W/m·K for the film growth at 1.3 V for the different deposition times of 5, 10, and 15 min, respectively. Taking into account the experimental error of the technique, all of these values are comparable. By comparing the obtained values with the values in the literature, it can be concluded that our values are similar for the material or even lower for the spongy film to those in the literature, which are 0.16 W/m·K or 0.19 W/m·K for PEDOT films grown by oxidative chemical vapor deposition (oCVD) [43] or electropolymerized PEDOT films using BTFMSI as the counter-ion [11], respectively. The thermal conductivity values measured in this study cannot be compared with the values of other PEDOT films electropolymerized using LiClO_4_ as a counter-ion, because they have not been measured until now.

Regarding the *zT* calculation, according to Cappai et al. [44], the anisotropy in the thermal conductivity depends on the PEDOT chain length. Because the PEDOT chain length is a difficult parameter to determine experimentally, because it is necessary to conduct the measurements in the liquid state and because PEDOT films cannot be dissolved in any solvent, it unclear whether the *zT* can be calculated. If the figure of merit at room temperature can be estimated in this particular case, it will be of (1.6 ± 0.6) × 10^−2^, (3.9 ± 1.0) × 10^−2^ and (2.8 ± 0.8) × 10^−2^ for films grown at 1.3 V and deposition times of 5, 10, and 15 min, respectively. The maximum value of the figure of merit calculated under previous assumptions is of (3.6 ± 0.6) × 10^−2^. This value is lower than the value (0.22) obtained by Culebras et al. [24] when BTFMSI was used as a counter-ion, but in our case, the process has a much lower cost. The power factor values measured in this study cannot be compared with the values of another PEDOT film electropolymerized using LiClO_4_ as a counter-ion, because they have not been measured until now. However, this is the first time that the figure of merit of electropolymerized PEDOT films was obtained in acetonitrile using lithium perchlorate as a counter-ion.

### 3.4. 3D-PEDOT Nanonetworks

To determine whether better-controlled nanostructuration and a controlled porosity confer an improvement on the thermoelectric properties of PEDOT, the material was electropolymerized inside one 3D AAO template. This 3D AAO template was prepared according to our previous work [29,30,34,45,46,47,48,49,50,51]. For that, the electropolymerization potential of 1.3 V was chosen because it yielded the maximum power factor in films. An electropolymerization time of 2 h was used, resulting in a thickness of 3.5 μm, as observed in the FE-SEM images of Figure 6A. The number of transversal channels was 15.

Once the 3D PEDOT nanostructures had been prepared, their thermoelectric properties, electrical conductivity, and Seebeck coefficient were measured. The electrical conductivity was calculated by using the sheet resistance ($R_{sheet}$), the number of the transversal channels (*N*), and the height of these transversal channels (*l*) (see Figure 6B), according to the study that was used for the 3D bismuth telluride nanonetwork [20], following Equation (2):(2)$$\sigma =\frac{1}{R_{sheet}\cdot N\cdot l}$$

The sheet resistance, the number of channels, and the transversal channels height were 1.2 × 10^2^ Ω/□, 15 nm, and 45 nm, respectively, yielding an electrical conductivity of 124 ± 6 S/cm. The Seebeck coefficient was found to be 19 ± 2 µV/K. The main advantage of growing these nanostructures was that they could be measured as normal films, without dissolving the AAO template. The power factor of the 3D-PEDOT nanostructures calculated from the electrical conductivity and Seebeck coefficient was 4.3 ± 0.9 µW/K·m^2^. On the basis of these findings, we can conclude that controlling the air gaps of the PEDOT spongy material yields no improvement in their thermoelectric performance. These results are different from what can be observed in inorganic materials such as 3D-Bi_2_Te_3_, where a strong improvement in the thermoelectric performance has been measured [19,20].

## 4. Conclusions

PEDOT films were grown using a low-cost technique based on EDOT electropolymerization in acetonitrile with lithium perchlorate as a counter-ion. The effect of the electropolymerization conditions (applied potential and deposition time) on the chemical structure and morphological properties of the films was investigated. It was shown that the vibrational modes of PEDOT films were not affected by the applied potential or the electropolymerization time, given that all the obtained films presented the typical vibrational modes of PEDOT. The morphology of PEDOT films depends on the applied potential. In the case of 1.3 V, a cauliflower structure was observed, and the size of these cauliflowers grew as the deposition time increased. In contrast, the morphology of PEDOT films grown at 1.4 V looked network-like. With regard to the Raman position of the bands, a blue shift was observed for most Raman modes as electropolymerization time increased independently of the applied potential. This behavior may be associated with the chemical bond length of molecules, due to changes in the structures induced by the growth conditions. The greatest Raman shift was observed in the symmetric C_α_ = C_β_ (-O) stretching band, the red shift of this vibrational mode may indicate that the chain in the resonant structure of PEDOT changed from a benzoid to a quinoid structure. The highest values for electrical conductivity, Seebeck coefficient, and power factor were found to be 158 ± 8 S/cm, 33 ± 1 µV/K, and 17 ± 2 µW/K·m^2^, respectively, for the PEDOT film grown at 1.3 V for 10 min. This power factor value was more than two times higher than the best values reported in the literature for electropolymerized PEDOT films in acetonitrile using lithium perchlorate as a counter-ion. The thermal conductivity of the film grown at 1.3 V for 10 min was estimated to be 0.13 ± 0.03 W/m·K. These resulted (in the case the PEDOT chains were not too long) in a figure of merit at room temperature of *zT* = (3.9 ± 1.0) × 10^−2^. In addition, 3D PEDOT nanostructures were electropolymerized inside 3D AAO, and their thermoelectric properties were measured; where they ended up being of the same order of magnitude as those of the PEDOT films. No further improvement was observed upon controlling the 3D nanostructuration; the opposite of what has been found in inorganic 3D networks was found instead. As a whole, an alternative approach to improve the thermoelectric performance of flexible and scalable PEDOT films was reported.

## Figures and Tables

**Figure 1 nanomaterials-12-04430-f001:**
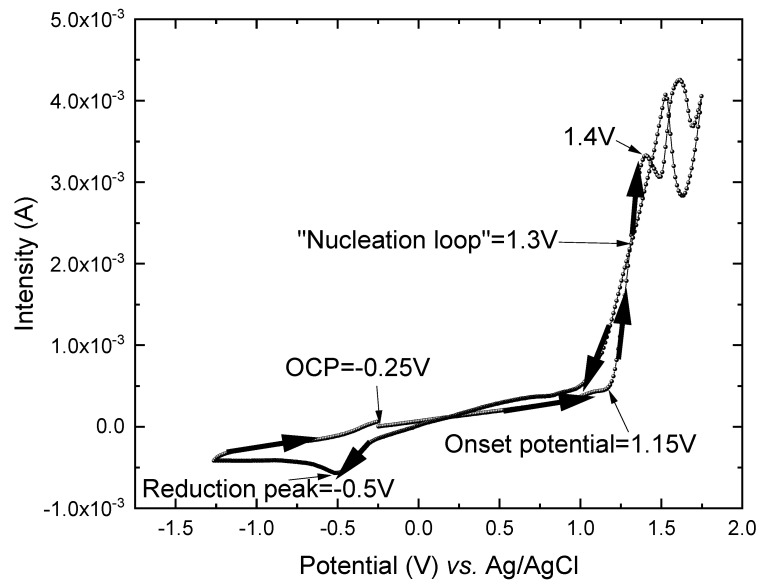
Cyclic voltammetry in acetonitrile of 0.01 M EDOT and 0.1 M LiClO_4_ with a scan rate of 0.01 V/s and at room temperature.

**Figure 2 nanomaterials-12-04430-f002:**
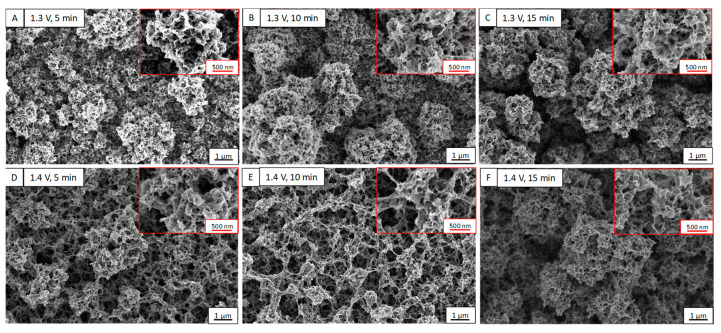
FE-SEM images of PEDOT films for different electropolymerization conditions, at (**A**) 1.3 V for 5 min, (**B**) 1.3 V for 10 min, (**C**) 1.3 V for 15 min, (**D**) 1.4 V for 5 min, (**E**) 1.4 V for 10 min, and (**F**) 1.4 V for 15 min.

**Figure 3 nanomaterials-12-04430-f003:**
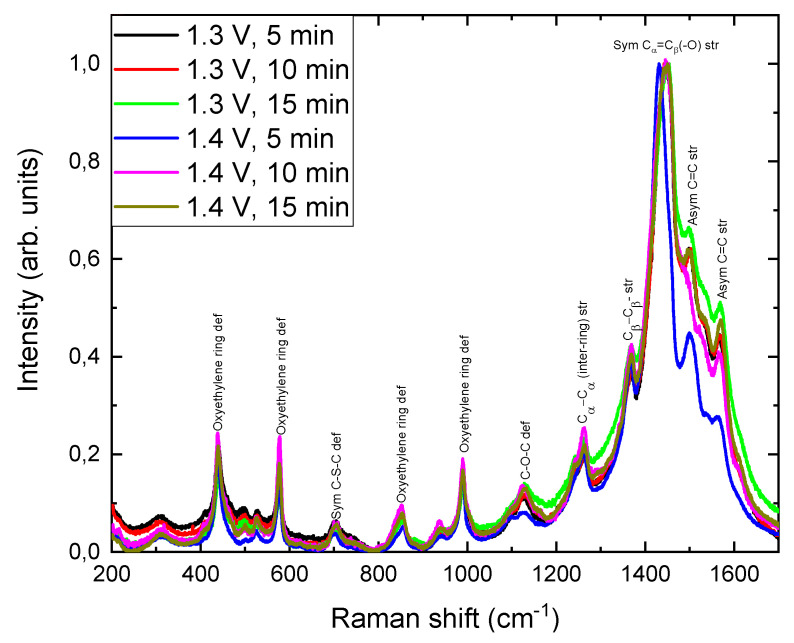
Raman spectra of the PEDOT films grown by electropolymerization under different oxidation potentials, i.e., 1.3 and 1.4 V, and electropolymerization times, i.e., 5, 10, and 15 min. Raman spectra were normalized to the band corresponding to the symmetric C_α_ = C_β_ (-O) stretching at 1445 cm^−1^.

**Figure 4 nanomaterials-12-04430-f004:**
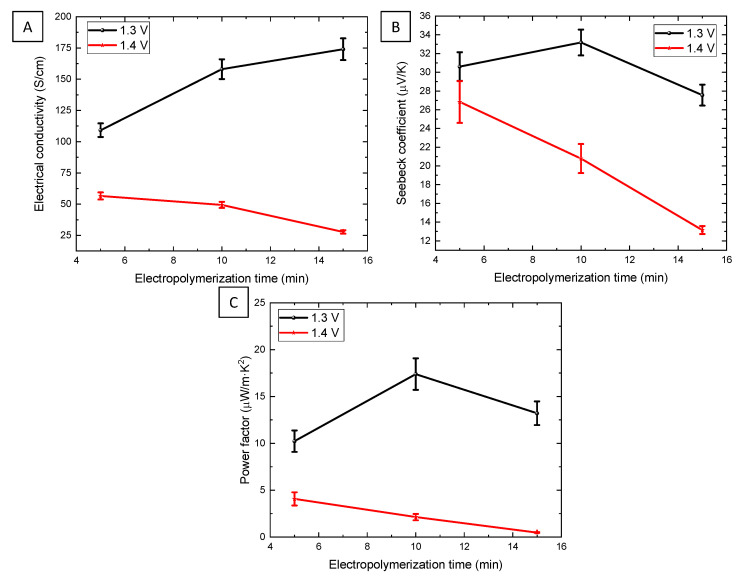
Thermoelectric properties: (**A**) the electrical conductivity, (**B**) the Seebeck coefficient, and (**C**) the power factor of PEDOT films grown at different potentials and deposition times.

**Figure 5 nanomaterials-12-04430-f005:**
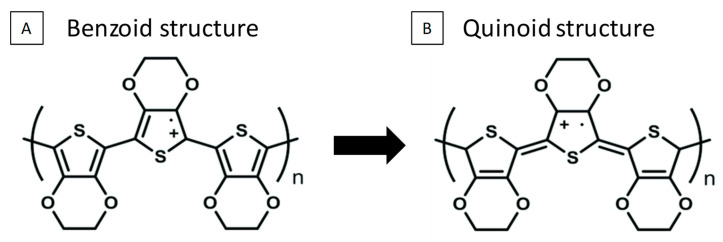
(**A**) benzoid structure and (**B**) quinoid structure.

**Figure 6 nanomaterials-12-04430-f006:**
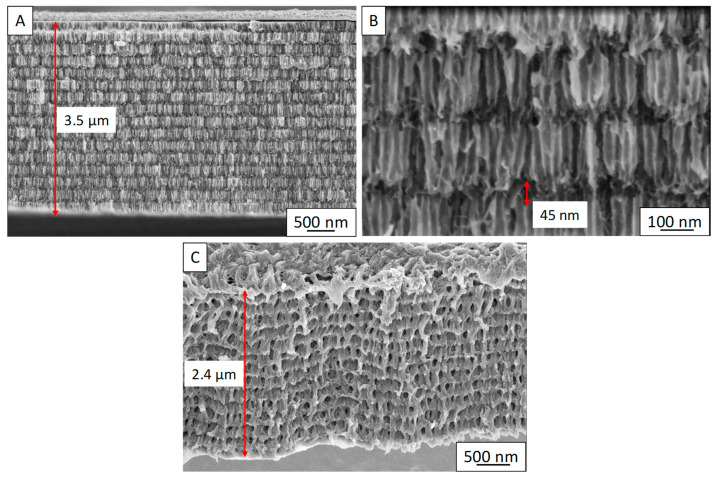
(**A**) FE-SEM of 3D PEDOT nanostructures inside AAO, (**B**) detail of the FE-SEM images of the transversal channels of the 3D PEDOT nanostructures inside AAO, and (**C**) freestanding 3D PEDOT nanostructures where the air gaps are controlled inside the PEDOT film.

**Table 1 nanomaterials-12-04430-t001:** Power factor of PEDOT:PSS films obtained by different solvents or treatments, as reported in the literature.

Different Solvents or Treatments	Power Factor (µW/m·K^2^)	Reference
Acetone	50	[6]
Sulfuric acid	50	[8]
Formic acid	80	[5]
At 150 °C	95	[7]
Ethylene glycol	470	[4]

**Table 2 nanomaterials-12-04430-t002:** Position, intensity, and the full width half maximum (FWHM) of the Raman vibration modes for the PEDOT films prepared by electropolymerization under different oxidation potentials, i.e., 1.3 and 1.4 V, and electropolymerization times, i.e., 5, 10, and 15 min.

Films		1.3 V, 5 min	1.3 V, 10 min	1.3 V, 15 min	1.4 V, 5 min	1.4 V, 10 min	1.4 V, 15 min
Thickness (µm)		4.1	6.0	7.3	4.3	10.4	23.2
Oxythel. Ring def	Position (cm^−1^)	440.7	441.1	441.2	439.6	440.3	441.1
	Intensity (arb. u.)	0.23	0.21	0.16	0.20	0.24	0.22
	FWHM (cm^−1^)	16.05	16.05	15.2	16.1	18.8	21.2
Oxythel. Ring def	Position (cm^−1^)	576.1	576.3	576.8	576.4	577.1	576.7
	Intensity (arb. u.)	0.18	0.15	0.13	0.13	0.24	0.18
	FWHM (cm^−1^)	12.7	12.3	12.6	12.8	11	11.5
Sym C-S-C def	Position (cm^−1^)	705	705	705	702	706	706
	Intensity (arb. u.)	0.064	0.052	0.047	0.038	0.053	0.057
	FWHM (cm^−1^)	45.3	35.7	28.8	27.1	27.8	30.5
Oxythel. Ring def	Position (cm^−1^)	849.9	852	851	852	848.4	850.4
	Intensity (arb. u.)	0.071	0.069	0.065	0.053	0.095	0.079
	FWHM (cm^−1^)	33.7	33.6	37	30.2	30.6	32.8
Oxythel. Ring def	Position (cm^−1^)	989.9	990.6	991	989	989.7	990.2
	Intensity (arb. u.)	0.15	0.14	0.14	0.17	0.19	0.17
	FWHM (cm^−1^)	17.5	17.6	17	12.6	16.1	15.7
C-O-C def	Position (cm^−1^)	1128	1130.9	1135.8	1123.7	1127.5	1129.6
	Intensity (arb. u.)	0.11	0.12	0.14	0.08	0.14	0.13
	FWHM (cm^−1^)	72.1	80	100	128	72.5	78.9
C_α_-C_α_	Position (cm^−1^)	1260.6	1261	1262.8	1257.5	1260.5	1261.2
	Intensity (arb. u.)	0.20	0.20	0.23	0.20	0.25	0.22
	FWHM (cm^−1^)	29	31	16.17	12.6	21.8	33.6
C_β_-C_β_	Position (cm^−1^)	1368.3	1369.3	1370.8	1367.6	1367.7	1369.1
	Intensity (arb. u.)	0.37	0.37	0.43	0.39	0.43	0.41
	FWHM (cm^−1^)	48.7	43.5	53	34.9	35.7	41.8
Sym. C_α_ = C_β_ (-O)	Position (cm^−1^)	1444.7	1444.7	1446.8	1432.9	1442.7	1444.2
	Intensity (arb. u.)	1	1	1	1	1	1
	FWHM (cm^−1^)	65.8	64.8	68.3	50.9	78.4	68.4
Asym. C = C_str_	Position (cm^−1^)	1496.1	1498.3	1494	1498.2	1486.7	1496.4
	Intensity (arb. u.)	0.6	0.6	0.65	0.44	0.55	0.6
	FWHM (cm^−1^)	152.5	181.1	103.7	77.9	72.7	52
Asym. C = C_str_	Position (cm^−1^)	1565.1	1566.2	1565.7	1559.6	1551.4	1566.3
	Intensity (arb. u.)	0.44	0.44	0.5	0.28	0.41	0.47
	FWHM (cm^−1^)	52	190	149.4	130.4	38.3	221.8

## Data Availability

Not applicable.

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
