# Peer review of "The Thermoelectric Properties of Spongy PEDOT Films and 3D-Nanonetworks by Electropolymerization"

_nanomaterials, 2022, doi:10.3390/nano12244430_

Round 1

Reviewer 1 Report

This work reported spongy PEDOT films by electropolymerization, and the electropolymerization conditions (applied potential and deposition time) of PEDOT films were investigated to improve their thermoelectric properties. The research significance of this work is unclear, and some comments need to be considered as follows:

Comment #1: The novelty of this work is less discussed. I suggest the authors compare with other similar previous reports, and demonstrate the research significance of this work in the Introduction.

Comment #2: The author claims that PEDOT film with 1.3V has a higher conductivity compared with 1.4V. Is it possible that the film made of 1.2V will have better electrical or thermoelectric properties? In addition, more evidence is necessary to confirm that the electropolymerization of PEDOT is reversible.

Comment #3: Raman spectra show that PEDOT films have a redshift from 1444.7 cm-1 to 1432.9cm-1 with increasing the applied potential. The quinoid structure of PEDOT is known to have better conductivity, so the electrical properties of PEDOT films with 1.4V should be better, which is contrary to the results of the article.

Comment #4: The conductivities of PEDOT films with 1.3V and 1.4V in this article show an opposite rule with the polymerization time, and the error range of Seebeck coefficient under the PEODT film with 5min at 1.3V is too large.

Comments #5: Please check the sentence and grammar in the article, such as “studying their thermoelectric properties obtaining lower values” in Abstract.

Author Response

Referee 1:

Comments to the Author: This work reported spongy PEDOT films by electropolymerization, and the electropolymerization conditions (applied potential and deposition time) of PEDOT films were investigated to improve their thermoelectric properties. The research significance of this work is unclear, and some comments need to be considered as follows:

 Comment #1: The novelty of this work is less discussed. I suggest the authors compare with other similar previous reports, and demonstrate the research significance of this work in the Introduction.

There are very few works dealing with the growth of PEDOT films by electropolymerization using the same counter ions used in our work, as stated in the introduction. That number of publications (three) is even lower when studying their thermoelectric properties. All the studies on these topics are included in the introduction, so the reader can understand the extent of research on this specific material for thermoelectric applications and the obtained values. Also, it is difficult to compare our results with most of the studies because, as the rest of the introduction explains, the thermoelectric properties are strongly dependent on the growth conditions.

Nevertheless, the significance of this work was stated at the end of the introduction, when it said that “this study opens a new approach in the fabrication of PEDOT films […]”. And, in the conclusions, when it said that “the obtained thermoelectric power factor obtained is twice the best reported in literature”. But to emphasize these statements, given the comment from the referee, we have changed these paragraphs to:

The last paragraph of the introduction: “This study opens a new approach in the fabrication of PEDOT films to improve the thermoelectric properties using a non-expensive and industrially scalable growth technique as electropolymerization for the fabrication of flexible thermoelectric devices“ was replaced by: “In our work, the influence of the potentiostatic mode in electropolymerized PEDOT films on thermoelectric properties was studied for the first time. Therefore, this work opens a new way to improve the thermoelectric properties of PEDOT films using a growth technique, which is non-expensive and industrially scalable, and that can be applied to the development of flexible thermoelectric devices.”

Also, a sentence was added at the end of the conclusions: “As a whole, an alternative approach to improve the thermoelectric performance of flexible and scalable PEDOT films has been reported.”    

 Comment #2: The author claims that PEDOT film with 1.3V has a higher conductivity compared with 1.4V. Is it possible that the film made of 1.2V will have better electrical or thermoelectric properties? In addition, more evidence is necessary to confirm that the electropolymerization of PEDOT is reversible.

We grew films at 1.2 V with different electrodeposition times, but no deposits were obtained because this voltage is placed before the nucleation loop in the cyclic voltammetry. Also, we did experiments at 1.5 V, and the electrical conductivity and Seebeck coefficient obtained in those films were very low, and this is the reason why we did not incorporate them in the manuscript. In the cyclic voltammetry, figure 1, it can be seen that the electropolymerization of PEDOT is reversible until 1.4 V, and therefore that value was chosen as the maximum electrodeposition potential applied. To clarify this, the following paragraph is included: “According to the cyclic voltammetry, 1.3 V and 1.4 V vs. Ag/AgCl were selected as oxidation potentials at the initial stage of nucleation processes and the final stage of the oxidation region, respectively, to grow and to tune the structural, morphological, and thermoelectric properties of PEDOT films. In addition, three electropolymerization times (5, 10, and 15 min) were also studied to know the influence of the film thicknesses on the thermoelectric properties.”  was replaced by: “According to the cyclic voltammetry, the potential region of interest ranged from 1.15 to 1.5 V vs. Ag/AgCl. Therefore, the electropolymerization potentials of 1.3 V and 1.4 V vs. Ag/AgCl were selected as the electropolymerization potentials at the initial stage of nucleation processes and the final stage of the oxidation region, respectively, to grow and tune the structural, morphological, and thermoelectric properties of PEDOT films. In addition, three electropolymerization times (5, 10, and 15 min) were also studied to know the influence of the film thicknesses on the thermoelectric properties. It is also worth to mentioning that no film was obtained when applying lower potentials, such as 1.2 V vs. Ag/AgCl, and for higher potentials, such as 1.5 V vs. Ag/AgCl the transport properties were too low to be considered of interest.”

We must also say that the electropolymerization of PEDOT is not reversible, as the cyclic voltammogram shows. We want to deposit PEDOT so we are looking for an irreversible process to obtain the film that we characterized. As a general rule, when you see these types of cyclic voltammogram you know the process is not reversible.

 Comment #3: Raman spectra show that PEDOT films have a redshift from 1444.7 cm-1 to 1432.9cm-1 with increasing the applied potential. The quinoid structure of PEDOT is known to have better conductivity, so the electrical properties of PEDOT films with 1.4V should be better, which is contrary to the results of the article.

The redshift observed by Raman is very low, which means that not much polymer has changed into the quinoid structure. Therefore, although there should be a certain percentage of the material modified, it is still not enough to have percolation of the quinoid structure to show higher electrical conductivity. So, in both cases, the conductivity is governed by the benzoid structure. We think that the reason for the higher conductivity of the films grown at 1.3 V is due to the presence of a more compact morphology than the films grown at 1.4 V, which means that the electrical conductivity should be higher in the case of 1.3 V. To clarify this point, the following sentence was added to the text: “, but in our case this change into quinoid is not enough to modify the whole polymer structure (the Raman shift is very low). Therefore, the electrical conductivity should not be affected by this structural change.”

Comment #4: The conductivities of PEDOT films with 1.3V and 1.4V in this article show an opposite rule with the polymerization time, and the error range of Seebeck coefficient under the PEODT film with 5min at 1.3V is too large.

The different behaviors in the electrical conductivity between the films grown at 1.3 V and 1.4 V are because the films grown at 1.3 V are more compact when the electropolymerization time is higher, and this does not happen in the case of the films grown at 1.4 V.

As for the error of the Seebeck coefficient and, in consequence, the error of the power factor, which has been checked and corrected in figure 4 and in the manuscript, it was a typo because it should be around 10% of the coefficient’s value. We would like to thank the reviewer for pointing that out.

 Comments #5: Please check the sentence and grammar in the article, such as “studying their thermoelectric properties obtaining lower values” in Abstract.

We have checked the grammar thought out the whole manuscripts. The last sentence in the abstract “studying their thermoelectric properties obtaining lower values.” was replaced by “obtaining lower values in their thermoelectric properties.”

Reviewer 2 Report

Thermoelectric properties of spongey PEDOT films obtained by potentiostatic electropolymerization were studied. The analysis of the experimental data was conducted well. The reviewer proposes that the work be published after minor revision.

The following are some comments and open questions:

1.       For Figure 1, if the scan rate is 0.01 V/s and consistent, why does the data point seem to have fewer in higher potential?

2.       This sentence may not be appropriate on line 286 since you previously said that "the values obtained in the literature for films produced in acetonitrile and lithium perchlorate were 200 S/cm." This research measured a maximum electrical conductivity of ~175 S/cm, which is just slightly higher than the Na:PSS in water (150 S/cm).

3.       Considering that the electrical conductivity of films deposited at 1,3 V is higher, why is the Seebeck coefficient likewise higher? In general, the higher the electrical conductivity, the lower the Seebeck coefficient will be. It would be better to offer a brief explanation.

4.       There is a typo in the unit in line 357. Please check it. 

Author Response

Dra. Cristina Vicente Manzano

E-mail: cristina.vicente@csic.es

Phone number: +34 91 806 0700,

Fax number: +34 91 806 0701

C/ Isaac Newton 8 PTM

Tres Cantos (Madrid)

Instituto de Micro y Nanotecnología (IMN)

Consejo Superior de Investigaciones Científicas (CSIC)

Madrid (Spain), 5th  Deceember 2022

Jiangjie Tian

Assistant Editor of Nanomaterials Manuscript ID: nanomaterials-2082657 Title: Thermoelectric Properties of Spongy PEDOT Films and 3D-Nanonetworks by

Electropolymerization

Dear Editor,

First of all, we want to thank the reviewers for their positive criticism and suggestions, as this has improved the quality and completeness of the manuscript. We have made the indicated technical corrections, which were proposed by the reviewers.

Please find below our answers to each of the reviewers' comments for our manuscript nanomaterials-2082657. Note that the reviewers' comments are written using italic fonts.

Sincerely,

Dr. Cristina Vicente Manzano and Prof. Marisol Martín-González

Referee 1:

Comments to the Author: This work reported spongy PEDOT films by electropolymerization, and the electropolymerization conditions (applied potential and deposition time) of PEDOT films were investigated to improve their thermoelectric properties. The research significance of this work is unclear, and some comments need to be considered as follows:

 Comment #1: The novelty of this work is less discussed. I suggest the authors compare with other similar previous reports, and demonstrate the research significance of this work in the Introduction.

There are very few works dealing with the growth of PEDOT films by electropolymerization using the same counter ions used in our work, as stated in the introduction. That number of publications (three) is even lower when studying their thermoelectric properties. All the studies on these topics are included in the introduction, so the reader can understand the extent of research on this specific material for thermoelectric applications and the obtained values. Also, it is difficult to compare our results with most of the studies because, as the rest of the introduction explains, the thermoelectric properties are strongly dependent on the growth conditions.

Nevertheless, the significance of this work was stated at the end of the introduction, when it said that “this study opens a new approach in the fabrication of PEDOT films […]”. And, in the conclusions, when it said that “the obtained thermoelectric power factor obtained is twice the best reported in literature”. But to emphasize these statements, given the comment from the referee, we have changed these paragraphs to:

The last paragraph of the introduction: “This study opens a new approach in the fabrication of PEDOT films to improve the thermoelectric properties using a non-expensive and industrially scalable growth technique as electropolymerization for the fabrication of flexible thermoelectric devices“ was replaced by: “In our work, the influence of the potentiostatic mode in electropolymerized PEDOT films on thermoelectric properties was studied for the first time. Therefore, this work opens a new way to improve the thermoelectric properties of PEDOT films using a growth technique, which is non-expensive and industrially scalable, and that can be applied to the development of flexible thermoelectric devices.”

Also, a sentence was added at the end of the conclusions: “As a whole, an alternative approach to improve the thermoelectric performance of flexible and scalable PEDOT films has been reported.”    

 Comment #2: The author claims that PEDOT film with 1.3V has a higher conductivity compared with 1.4V. Is it possible that the film made of 1.2V will have better electrical or thermoelectric properties? In addition, more evidence is necessary to confirm that the electropolymerization of PEDOT is reversible.

We grew films at 1.2 V with different electrodeposition times, but no deposits were obtained because this voltage is placed before the nucleation loop in the cyclic voltammetry. Also, we did experiments at 1.5 V, and the electrical conductivity and Seebeck coefficient obtained in those films were very low, and this is the reason why we did not incorporate them in the manuscript. In the cyclic voltammetry, figure 1, it can be seen that the electropolymerization of PEDOT is reversible until 1.4 V, and therefore that value was chosen as the maximum electrodeposition potential applied. To clarify this, the following paragraph is included: “According to the cyclic voltammetry, 1.3 V and 1.4 V vs. Ag/AgCl were selected as oxidation potentials at the initial stage of nucleation processes and the final stage of the oxidation region, respectively, to grow and to tune the structural, morphological, and thermoelectric properties of PEDOT films. In addition, three electropolymerization times (5, 10, and 15 min) were also studied to know the influence of the film thicknesses on the thermoelectric properties.”  was replaced by: “According to the cyclic voltammetry, the potential region of interest ranged from 1.15 to 1.5 V vs. Ag/AgCl. Therefore, the electropolymerization potentials of 1.3 V and 1.4 V vs. Ag/AgCl were selected as the electropolymerization potentials at the initial stage of nucleation processes and the final stage of the oxidation region, respectively, to grow and tune the structural, morphological, and thermoelectric properties of PEDOT films. In addition, three electropolymerization times (5, 10, and 15 min) were also studied to know the influence of the film thicknesses on the thermoelectric properties. It is also worth to mentioning that no film was obtained when applying lower potentials, such as 1.2 V vs. Ag/AgCl, and for higher potentials, such as 1.5 V vs. Ag/AgCl the transport properties were too low to be considered of interest.”

We must also say that the electropolymerization of PEDOT is not reversible, as the cyclic voltammogram shows. We want to deposit PEDOT so we are looking for an irreversible process to obtain the film that we characterized. As a general rule, when you see these types of cyclic voltammogram you know the process is not reversible.

 Comment #3: Raman spectra show that PEDOT films have a redshift from 1444.7 cm-1 to 1432.9cm-1 with increasing the applied potential. The quinoid structure of PEDOT is known to have better conductivity, so the electrical properties of PEDOT films with 1.4V should be better, which is contrary to the results of the article.

The redshift observed by Raman is very low, which means that not much polymer has changed into the quinoid structure. Therefore, although there should be a certain percentage of the material modified, it is still not enough to have percolation of the quinoid structure to show higher electrical conductivity. So, in both cases, the conductivity is governed by the benzoid structure. We think that the reason for the higher conductivity of the films grown at 1.3 V is due to the presence of a more compact morphology than the films grown at 1.4 V, which means that the electrical conductivity should be higher in the case of 1.3 V. To clarify this point, the following sentence was added to the text: “, but in our case this change into quinoid is not enough to modify the whole polymer structure (the Raman shift is very low). Therefore, the electrical conductivity should not be affected by this structural change.”

Comment #4: The conductivities of PEDOT films with 1.3V and 1.4V in this article show an opposite rule with the polymerization time, and the error range of Seebeck coefficient under the PEODT film with 5min at 1.3V is too large.

The different behaviors in the electrical conductivity between the films grown at 1.3 V and 1.4 V are because the films grown at 1.3 V are more compact when the electropolymerization time is higher, and this does not happen in the case of the films grown at 1.4 V.

As for the error of the Seebeck coefficient and, in consequence, the error of the power factor, which has been checked and corrected in figure 4 and in the manuscript, it was a typo because it should be around 10% of the coefficient’s value. We would like to thank the reviewer for pointing that out.

 Comments #5: Please check the sentence and grammar in the article, such as “studying their thermoelectric properties obtaining lower values” in Abstract.

We have checked the grammar thought out the whole manuscripts. The last sentence in the abstract “studying their thermoelectric properties obtaining lower values.” was replaced by “obtaining lower values in their thermoelectric properties.”

Referee 2:

 Comments to the Author: Thermoelectric properties of spongey PEDOT films obtained by potentiostatic electropolymerization were studied. The analysis of the experimental data was conducted well. The reviewer proposes that the work be published after minor revision.

The following are some comments and open questions:

  1. For Figure 1, if the scan rate is 0.01 V/s and consistent, why does the data point seem to have fewer in higher potential?

The scan rate is constant in full cyclic voltammogram. The scan rate does not fix the number of points; it fixes the scan rate. If there are fewer points, it is because there are more processes happening at higher potential.

  1. This sentence may not be appropriate on line 286 since you previously said that "the values obtained in the literature for films produced in acetonitrile and lithium perchlorate were 200 S/cm." This research measured a maximum electrical conductivity of ~175 S/cm, which is just slightly higher than the Na:PSS in water (150 S/cm).

The referee is right, it is not appropriate and so the sentence was substituted for "Then, the maximum electrical conductivity obtained in this study was in the order of magnitude of the values found in the literature."

  1. Considering that the electrical conductivity of films deposited at 1,3 V is higher, why is the Seebeck coefficient likewise higher? In general, the higher the electrical conductivity, the lower the Seebeck coefficient will be. It would be better to offer a brief explanation.

The film with the higher electrical conductivity was grown at 1.3 V for 15 minutes and this film has the lower Seebeck coefficient of all the films grown at 1.3 V. So, it seems to follow the trend that the referee says. Now, if we compare the properties between the films grown at 1.3 V and 1.4 V, we see that both the transport properties, electrical conductivity and Seebeck coefficient, are different, but we cannot compare them directly because their morphologies are different in terms of porosity. The porosity has a strong influence on the electrical conductivity but not in the Seebeck,

  1. There is a typo in the unit in line 357. Please check it. 

The unit was corrected.

Round 2

Reviewer 1 Report

I think this manuscript can be accepted.